# Differences in Driver Behavior between Manual and Automatic Turning of an Inverted Pendulum Vehicle

**DOI:** 10.3390/s22249931

**Published:** 2022-12-16

**Authors:** Chihiro Nakagawa, Seiya Yamada, Daichi Hirata, Atsuhiko Shintani

**Affiliations:** Mechanical Engineering Department, Osaka Metropolitan University, 1-1 Gakuen-cho, Naka, Sakai 599-8531, Osaka, Japan

**Keywords:** personal mobility vehicle, inverted pendulum vehicle, automatic driving, dynamics of a driver, joint moment

## Abstract

Personal mobility vehicles (PMVs) are compact and lightweight compared to automobiles; hence, human dynamic behavior affects a vehicle’s postural stability. In this study, the dynamic behaviors of drivers of inverted pendulum vehicles (IPV) under manual and automatic driving were investigated. One particular feature of applying automatic driving to IPV is constant posture stabilization control. In this study, the drivers’ center of gravity (COG)/center of foot pressure position (COP) and joint moments during turning were investigated experimentally. It was found that the drivers’ COG shifted backward during turning and deceleration. For COP, it was found that drivers maintained balance by moving their inner foot more inward and their outer foot more outward during turning. These results are significant for understanding the steps taken to withstand centrifugal forces during turning. The joint moments of the foot were more significant in automatic turning than in manual turning to prevent falling owing to centrifugal force. These findings can facilitate the development of an automatic control method that shifts the COG of a driver, as in manual turning.

## 1. Introduction

Recently, research on automated driving technology for automobiles has rapidly developed [1,2]. Studies are also underway in anticipation of a mobile society in which automatic driving vehicles will be widely used and vehicles will communicate with each other [3,4,5]. Research on automatic driving includes studies on scene classification [6] and driver monitoring [7]; studies focusing on humans include motion sickness [8,9,10]. However, automatic driving, focusing on humans riding in an inverted pendulum vehicle (IPV) as a compact standing personal mobility vehicle (PMV), has rarely been studied. Unlike automobiles, human dynamic behavior critically affects the dynamic stability of PMV systems. The objective and motivation of this research were not focused on system security, but sought to improve dynamic stability, and to safely apply automated driving to PMVs, which are personal transportation vehicles [11,12,13]. An IPV is a representative PMV [14]. In an IPV, the driver rides on a step between two wheels and moves by shifting the driver’s center of gravity [15,16]. The vehicle turns and can be used in a small space by changing the rotational speed of the right and left wheels. However, when an IPV is used in a crowded space, there is a risk of collision with pedestrians or obstacles, particularly when the driver’s attention is diverted. Automatic driving is expected to ensure the safety of drivers and pedestrians in situations in which the driver’s perception is problematic. One of the problems in applying automatic driving to an IPV is that the vehicle requires constant stabilization control. During automatic driving, stabilization control may be substantially affected depending on the input value to the wheels, resulting in the risk of disturbing the driver’s posture. Furthermore, since the vehicle is compact and lightweight, the mass of the driver relative to that of the vehicle is large, and the behavior of the driver has a significant impact on the entire system. Therefore, it is important to understand the mechanism of driver behavior during automatic operation to apply it to an IPV safely. We demonstrated the effect of automatic braking on an IPV [17]. In this study, we experimentally investigated differences in driver behavior between manual and automatic turning. Showing this difference is the contribution of this study, because knowing the difference allows us to examine the guidelines for automatic driving.

The outline of this paper is as follows. In Section 2, we present the experimental vehicle and automatic turning system. In Section 3, we present the experiment carried out to derive COG and COP of the drivers. We analyze the results. Based on the analysis, additional experiments are carried out in Section 4 to derive the joint moments in order to further understand the details of human motion. In Section 5, the conclusions of this study and future issues are presented.

## 2. Experimental Vehicle and the Automatic Turning System

An IPV (FIT Co., Ltd., Gifu, Japan) was used as the experimental vehicle, as shown in Figure 1. Table 1 lists the specifications of the analysis.

An IPV mainly consists of a step, a handle, and two wheels. The wheels were parallel to each other on the left and right sides of the vehicle and were mounted on a step platform, each with a motor.
(1)$$T_{R}=T_{L}=K_{1}\theta_{vehicle}+K_{2}\omega_{vehicle}$$

Equation (1) is the control equation describing a vehicle moving straight ahead [18]. The step was equipped with a gyro sensor, which fed back the tilt angle $\theta_{vehicle}$ and angular velocity $\omega_{vehicle}$ of the step, and applied torques $T_{R}$ and $T_{L}$ equally to the right and left wheels to enable stabilization control and straight driving. Note that $K_{1}$ and $K_{2}$ represent the feedback gains defined in Table 2, and $T_{R}$, $T_{L}$, $\theta_{vehicle}$, and $\omega_{vehicle}$ follow the definitions shown in Figure 2.
(2)$$T_{R}=K_{1}\theta_{vehicle}+K_{2}\omega_{vehicle}+K_{3}\theta_{handle},$$
(3)$$T_{L}=K_{1}\theta_{vehicle}+K_{2}\omega_{vehicle}-K_{3}\theta_{handle}$$

Equations (2) and (3) represent the control equations for manual turning. The two wheels had a steering angle of zero, and the vehicle could turn by varying the output of the motors attached to each wheel from left to right. The handle was located at the front center of the vehicle, attached to the step, and a rotation angle sensor was attached to the connection. When the handle was steered, in addition to the straight driving operation, the step tilt angle $\theta_{vehicle}$, angular velocity $\omega_{vehicle}$, and handle angle $\theta_{handle}$ were fed back, and the control torque $T_{R}$ was input to the right wheel and $T_{L}$ to the left wheel, thereby stabilizing the control and enabling manual turning. Note that $K_{1}$, $K_{2}$, and $K_{3}$ represent the feedback gains defined in Table 2, and $T_{R}$, $T_{L}$, $\theta_{vehicle}$, $\omega_{vehicle}$, and $\theta_{handle}$ follow the definitions shown in Figure 2.
(4)$$T_{R}=K_{1}\theta_{vehicle}+K_{2}\omega_{vehicle}+\alpha$$
(5)$$T_{L}=K_{1}\theta_{vehicle}+K_{2}\omega_{vehicle}-\alpha$$

Equations (4) and (5) are the control equations for the automatic turning. These equations were obtained by replacing the third term on the right side of Equations (2) and (3), $K_{3}\theta_{handle}$, with a constant α, which is independent of the handle angle $\theta_{handle}$ and is arbitrarily determined in this study based on the experimental conditions. When the vehicle was operated in a straight line, the step tilt angle $\theta_{vehicle}$ and angular velocity $\omega_{vehicle}$ were fed back, and the control torque $T_{R}$ was input to the right wheel and $T_{L}$ to the left wheel, enabling stabilization control and automatic turning. In other words, because $\theta_{handle}$ is not fed back, even if the driver operates the handle, the steering intention is not transmitted to the vehicle, and the vehicle turns without the handle tilting in the turning direction. Note that $K_{1}$, $K_{2}$, and $K_{3}$ represent the feedback gains defined in Table 2, and $T_{R}$, $T_{L}$, $\theta_{vehicle}$, and $\omega_{vehicle}$ follow the definitions shown in Figure 2. $K_{1}$, $K_{2}$, and $K_{3}$ were arbitrarily determined in this study such that the IPV could maintain postural stability.

## 3. Derivation Experiment for COG/COP

Manual and automatic turning experiments were performed using the vehicle described in Section 2. The center of gravity (COG) of the driver plays an important role in the stability of the system, including the stability of the vehicle and the driver. Therefore, an experimental methodology was designed to derive the COG of the driver. In addition to COG, the center of foot pressure position (COP) should also be determined because the stepping of the left and right legs is crucial for turning.

### 3.1. Experimental Methods and Conditions

Three healthy young men (aged 24–25 years, with an average height of 173.8 cm and an average weight of 68.6 kg) participated in the experiment. Before the experiment, the drivers practiced driving the experimental vehicle manually for 30 min to become sufficiently familiar with riding and steering the vehicle to minimize the differences in driving skills between the subjects.

The experiment was conducted under two conditions: manual turning by the driver, and automatic turning. The former condition was achieved by tilting the handle to the maximum in a specified direction. The latter condition was achieved by activating the automatic turning system with a random direction and timing, including a condition in which the vehicle did not turn, making it difficult for the driver to predict the turning direction. This increases the reproducibility of the automatic operation. The turning time was set to 1.5 s. For each condition, the vehicle speed was set as either low or high. The results were 3.3 ± 0.3 km/h for low speed and 4.9 ± 0.3 km/h for high speed. For each condition, the turning sensitivity was varied in two steps: The turning sensitivity was achieved by changing the value of $\theta_{handle}$ in Equations (2) and (3). Three sets of experimental data were obtained for each of the eight experimental conditions. Data that could not be analyzed due to a large amount of missing data were excluded from the analysis.

The time history of the turning experiment is shown in Figure 3. This is the case for automatic turning, where the vehicle velocity is low and turning sensitivity is high. For each section, (a) (−1 s–0 s) is before turning, (b) (0 s) is at the beginning of turning, (c) (0 s–1.5 s) occurs during turning, and (d) (1.5 s) occurs at the end of turning. For reference, the data are shown in Figure 4 in correspondence with the acquired analysis data described below. The evaluation was conducted by measuring the displacements of the driver’s COG and COP during turning relative to the stationary state for each condition.

### 3.2. Measurement System

In this experiment, the position of each part of the human body was measured using 12 cameras in the 3D motion analysis system Flex3 (OptiTrack) (Figure 5a). The 3D motion analysis system can be used to analyze detailed human movements [19]. The sampling rate was set to 100 Hz. Markers were attached to the driver at 39 points based on a Plug-in Gait full-body model. The position data output from the 3D motion analysis system was based on a fixed coordinate system on the ground; therefore, it was converted to a fixed coordinate system on the vehicle. Hence, the position and yaw angle of the vehicle were measured by attaching markers to the vehicle at four points, and the coordinates were converted. The positions of the markers on the vehicle are shown in Figure 1. The fixed vehicle coordinate system was a right-handed Cartesian coordinate system with the *x*-axis in the traveling direction of the vehicle, *y*-axis in the lateral direction, and *z*-axis in the direction that formed a right-handed system with the *x*- and *y*-axes. The COP, on the contrary, was measured using the floor reaction force sensor M3D-EL-FP-U-C2005 (Tech-Gihan Co., Ltd., Kyoto, Japan) (Figure 5b), which can be attached to a human foot to measure the reaction force of one foot at a time.

### 3.3. Analysis Methods

#### 3.3.1. COG

In this experiment, we used the center of gravity (COG) and center of pressure (COP), which is the position of the reaction force applied to both feet, as indices to evaluate the behavior of the driver.
(6)$$r_{COG}=\frac{\sum m_{i}r_{i}}{\sum m_{i}}$$

$r_{COG},$ which is the position of COG, was calculated from the mass $m_{i}$ of each body part and the position $r_{i}$ of the center of mass using Equation (6). The mass and center of mass of each body part were derived using the body part coefficients presented in [20].

#### 3.3.2. Centrifugal Force

Centrifugal force contributes to a driver’s postural stability. In this study, the centrifugal force applied to the vehicle was used to evaluate the behavior of the driver, because only the value determined by the vehicle was used in the discussion, eliminating the human parameter (body weight). The method proposed by the authors for calculating the centrifugal force applied to a vehicle is as follows. The equation is derived from the fact that multiplying the radius by the angular velocity yields the circumferential velocity.
(7)$$v_{R}=\left(r+d\right)\omega$$
(8)$$v_{L}=\left(r-d\right)\omega$$

We assumed that the vehicle turned left, as shown in Figure 6. If the angular velocity is ω, turning radius is r, and distance between the two wheels is 2d during the turn, the velocities $v_{R}$ and $v_{L}$ of the left and right wheels can be expressed as in Equations (7) and (8).
(9)$$\omega =\frac{\left(v_{R}-v_{L}\right)}{2d}$$

Solving Equations (7) and (8) for ω yields Equation (9).
(10)$$v=\frac{\left(v_{R}+v_{L}\right)}{2}$$

The vehicle velocity v is described in Equation (10).
(11)$$F_{cent}=mr\omega^{2}$$

The centrifugal force $F_{cent}$ acting on the driver is described by Equation (11), where ω is the angular velocity, r is the turning radius, and m is the mass of the vehicle.
(12)$$F_{cent}=m\frac{\left(v_{R}-v_{L}\right)\left(v_{R}+v_{L}\right)}{4d}$$

Using the equation v=rω and Equation (11), $F_{cent}$ can be expressed as in Equation (12): the right turn can be written in the same manner by replacing $v_{R}$ and $v_{L}$ in the previous equation.

#### 3.3.3. Strength of Correlation

The strength of the correlation was evaluated according to a previous study [21]. Table 3 presents the correlation coefficients and classification of the strength of the correlations.

### 3.4. Experimental Results

First, we examined whether the COG (*x*-axis) displacement of the driver before and after turning differed between manual and automatic driving conditions. The average values of COG (*x*-axis) displacement before and after turning are shown in Figure 7. In Figure 7, the inward direction of the turn is the positive *y*-axis (the blue graph is the result of manual turning, and the red graph is the result of automatic turning). The results show that the mean displacement of the COG (*x*-axis) during manual operation was 0.069 m (SE = 0.006) before turning and 0.017 m (SE = 0.007) after turning. Welch’s *t*-test [22] showed a significant difference (**) in COG (*x*-axis) displacement before and after turning during manual operations (t (43) = 5.368, *p* < 0.01). The mean COG (*x*-axis) displacement during automatic driving was 0.046 m (SE = 0.005) before turning and −0.001 m (SE = 0.008) after turning. Welch’s *t*-test showed a significant difference (**) in the COG (*x*-axis) displacement before and after turning during automatic driving (t (44) = 5.060, *p* < 0.01). In other words, the COG moved backward more significantly during turning than before turning, resulting in a deceleration.

Next, we examined whether the driver’s COG (*y*-axis) displacement differed between manual and automatic driving both before and after turning. The average values of COG (*x*-axis) displacement before and after turning are shown in Figure 8. In Figure 8, the inward direction of the turn is the positive *y*-axis (the blue graph is the result of manual turning, and the red graph is the result of automatic turning). The results show that the mean displacement of the COG (*y*-axis) before turning was 0.001 m (SE = 0.004) for manual driving and −0.001 m (SE = 0.003) for automatic driving. Welch’s *t*-test showed no significant difference (n.s.) in COG (*y*-axis) displacement among the driving modes before turning (t (44) = 0.472, n.s.). The mean COG (*y*-axis) displacement after turning was 0.028 m (SE = 0.005) for manual driving and −0.015 m (SE = 0.005) for automatic driving. Welch’s *t*-test showed a significant difference (**) in COG (*y*-axis) displacement between the driving modes after turning (t (48) = 5.719, *p* < 0.01). That is, the center of gravity shifted inward during manual turning and it significantly shifted outward during automatic turning. This may be due to the centrifugal force applied to the driver during automatic turning.

To examine the effect of centrifugal force on the lateral attitude stability of the driver during a turn, the relationship between the average value of centrifugal force and the maximum value of COG (*y*-axis) displacement during turning for the manual and automatic turning conditions was determined, as shown in Figure 9 and Figure 10. The data indicated with red dots in Figure 10 represent the fallen condition. The correlation coefficients for each scatter plot were −0.004 and 0.378 for manual and automatic turning, respectively, indicating a weak positive correlation between the centrifugal force and the maximum COG (*y*-axis) displacement for automatic turning. The lack of correlation in manual turning can be attributed to a larger COG (*y*-axis) displacement caused by turning the handle in the roll direction inside the turn than the effect of the centrifugal force. This indicates that the driver is easily affected by centrifugal force during automatic turning, and that there is a risk of falling when the centrifugal force acting on the vehicle exceeds approximately 50 N.

The mean value of the maximum displacement of the COG (*y*-axis) for each driving style was calculated to examine the extent to which the driving mode (manual or automatic) affects the lateral posture stability of the driver. The results were 0.089 m (SE = 0.009) and 0.043 m (SE = 0.003) for manual and automatic driving, respectively. A Welch’s *t*-test showed that displacement was significantly greater for manual turning (t (29) = 4.742, *p* < 0.01). This may be due to the fact that in manual turning, the entire body tilted significantly in the roll direction when the handle was rotated in that direction.

Figure 11 shows the average COP (*y*-axis) values for both feet during turning, which indicate the effects of the position of the feet on the lateral postural stability of the drivers. In Figure 11, the positive *y*-axis represents the inward direction of the turn. The COP of the inner leg during manual operation was 4.459 mm (SE = 0.995) and that of the outer leg was −9.342 mm (SE = 1.270). In automatic driving, the values were 1.461 mm (SE = 1.002) for the inward turning of the leg and −8.627 mm (SE = 1.150) for the outward turning of the leg. A Welch’s *t*-test showed a significant difference (**) between the COP (*y*-axis) of the foot on the inside of the turn and the COP (*y*-axis) of the foot on the outside of the turn, regardless of the driving style (manual driving, t (42) = 8.367, *p* < 0.01) (automatic driving, t (51) = 6.490, *p* < 0.01). In other words, it was found that during turning, the foot inside the turn moved more inward and the foot outside the turn moved more outward. This could be attributed to the axle foot shifting inward during turning, with the other foot being used to maintain balance.

## 4. Derivation Experiment for Joint Moments

In the previous section, the dynamic behavior of the driver was characterized in terms of the COG and COP. In fact, there are other ways to characterize the dynamic behavior of human. For example, the method to construct a human model and evaluate the internal load of humans through simulation [23,24], the method to reproduce dynamic behavior by using anthropomorphic bipedal robot [25], and the method to evaluate muscle strength by estimating joint moments [26]. Of all the methods, we focused on the method evaluating joint moment. When humans attempt to maintain balance, they stabilize their overall posture by moving their muscles and applying appropriate moments to their joints throughout their body. These moments are called joint moments. Methodologically, estimating the joint moment reveals where a human generates a force. The following section discusses the results of additional experiments focused on drivers’ joint moments. We can understand how drivers move their legs during manual and automatic turns by estimating the joint moments.

### 4.1. Additional Experimental Methods and Conditions

Additional experiments were conducted on five healthy young men (average height, 172.4 cm; average weight, 56.24 kg). These participants were completely different from those who participated in the experiment described in Section 3.1. Before the experiment was conducted, the drivers practiced driving the experimental vehicle manually for 30 min to become fully accustomed to riding and steering the vehicle, and to minimize differences in driving skills among the participants.

The experimental conditions consisted of two driving methods (manual and automatic), three turning directions (left turn, right turn, and straight ahead), two turning speeds (low speed (2 km/h) and high speed (6 km/h)), two turning radius (0.5 and 1.5 m) for the low speed, and two turning radius (1.5 and 3.0 m) for high speed. Two measurements were performed for each condition, resulting in a total of 40 measurements per participant. The experiment was conducted by driving a vehicle on a specified course at a specified speed under manual turning conditions. In the automatic turning condition, the vehicle was driven on an arbitrary course with the automatic turning system activated while driving straight ahead at a specified speed. All experimental conditions were randomized to make it difficult for the driver to predict the direction of the turn in the automatic driving condition, thereby improving the reproducibility of automatic driving. When the driver was requested to stop during the experiment, only the data prior to the interruption were analyzed. In addition, data that could not be analyzed because of large data deficits were excluded from the analysis.

Figure 12 shows the flow diagram of the experimental setup. The experimental data were obtained by quantitatively measuring the driver’s behavior and recording data on a personal computer (PC) and data logger.

### 4.2. Joint Moment Estimation System

Inverse kinematics and dynamics analyses were conducted to estimate the joint moments. Equations (13) and (14), which represent the general equations of inverse dynamics and consider the variables considered in this study, were used for the calculations. The definitions of the main variables are listed in Abbreviations and a summary diagram is presented in Figure 13.
(13)mp¨W+ω˙˜WRWc+ω˜2WRWc=mg+∑finternal+∑fexternal
(14)IWpω˙W+ωW×IWpωW+cW−pW×mp¨W=cW−pW×mg+∑tinternal+∑texternal

As shown, external forces on the driver’s hands and feet are required to estimate joint moments. In addition to the experimental vehicle used in the turning experiment described in Section 3, the handle reaction force meter (TEL-1KN-29, Tensar Co., Ltd., Blackburn, UK), shown in Figure 14, was attached to the experimental vehicle to measure all external forces applied to the driver. Inverse kinematic and inverse dynamic analyses were performed using the DhaibaWorks software to estimate the joint moments at each joint of the driver.

### 4.3. Analysis Method

Figure 15 shows the data processing flow used to determine joint moments. The inverse kinematics analysis used to calculate joint moments required external forces based on an absolute coordinate system. However, the values measured by the measurement equipment were based on the coordinate system of the equipment. In this section, the definitions of the main coordinate systems and the calculation method of the external force are presented.

#### 4.3.1. Definition of Coordinate Systems

In the absolute coordinate system, the origin was fixed to the ground at the starting point of the turn in manual operation, where the z-axis is the forward direction, the y-axis is the vertical upward direction, and the x-axis is the horizontal direction, forming a right-hand system with the y- and z-axes. Figure 16 shows the absolute coordinate system.

In the vehicle coordinate system, the origin was fixed to the step, where the $Y_{V}$-axis is the horizontal direction in front of the vehicle when the driver was riding in the vehicle and maintaining a stationary state, the $Z_{V}$-, $Y_{V}$-, and $Z_{V}$-axes are all the vertical upward direction, and the $X_{V}$-axis is the horizontal direction facing the vehicle sideways, which was on the right-hand side of the system. Figure 17 shows the vehicle coordinate system.

In the wearable floor reaction force sensor coordinate system, the origin was fixed to the left and right wearable floor reaction force sensors, where the $Y_{W}$-axis is the forward direction of the device when attached to the driver, the $Z_{W}$-axis iz the vertical downward direction of the device in a similar case, and the $X_{W}$-axis is the lateral direction of the device, which is the right-handed coordinate system with $Y_{W}$- and $Z_{W}$-axes. Figure 18 shows the coordinate system of the wearable floor reaction force sensor.

In the coordinate system of the handle reaction force sensor, the origin was fixed to the left and right handle reaction force sensors, where the $X_{h}$-axis is the backward direction of the vehicle when attached to the experimental vehicle, the $Z_{h}$-axis is the vertical upward direction of the vehicle in a similar case, and the $Y_{h}$-axis is the lateral direction of the vehicle, which is the right-handed coordinate system with the $X_{h}$- and $Z_{h}$-axes. Figure 19 shows the coordinate system of the handle reaction force sensor.

#### 4.3.2. Calculation of External Force Using the Measuring Devices

Two wearable floor reaction force sensors were attached to the left and right feet of each driver, respectively. The right wearable floor reaction force sensor measured the *X*-, *Y*-, and *Z*-component forces (i.e., $F_{foot\_R\_X}$, $F_{foot\_R\_Y}$, and $F_{foot\_R\_Z}$, respectively) applied to the main body of the device in the wearable floor reaction force sensor coordinate system. Conversely, the left wearable floor reaction force sensor measured the *X*-, *Y*-, and *Z*-component forces (i.e., $F_{foot\_L\_X},\;F_{foot\_L\_Y}$, and $F_{foot\_L\_Z},$ respectively) applied to the main body of the device in the wearable floor reaction force sensor coordinate system. Therefore, based on the wearable floor reaction force measurement coordinate system described in the previous section, the *X-*, *Y-*, and *Z*-components of the right sole reaction force were $-F_{foot\_R\_X}$, $-F_{foot\_R\_Y},\;\mathrm{and}\;-F_{foot\_R\_Z}$, respectively, and the *X-*, *Y-*, and *Z*-components of the left sole reaction force were $-F_{foot\_L\_X}$, $-F_{foot\_L\_Y},\;\mathrm{and}-F_{foot\_L\_Z}$, respectively.

Two handle reaction force sensors were attached to the left and right handle bars. The right-hand handle reaction force sensor measured the force $F_{hand\_R}$ applied to the device in the front–back direction, whereas the left-hand handle reaction force sensor measured the force $F_{hand\_L}$ applied to the device in the left–right direction. Therefore, the handle reaction forces applied to the driver’s left and right hands were $-F_{hand\_R}$, $-F_{hand\_L}$, and 0 for the *X-*, *Y-*, and *Z*-components, respectively.

#### 4.3.3. Human Dynamics Model

The heights and weights of the participants were measured beforehand and a human dynamics model was developed based on this information. The human dynamics model was constructed using DhaibaWorks [27], which was based on a database of 6700 Japanese people at the Research Institute of Human Engineering for Quality Life, and was derived from a principal component analysis of their body dimensions and overall shape. The model consisted of 20 rigid bodies (pelvis, spine, sternum, neck, clavicles, humeri, ulnas, hands, femurs, tibias, feet, and toes) connected by 20 joints. Each joint was rotated around the three axes. Figure 20 shows the configuration of the human dynamics model, and Table 4 and Table 5 list the names of bones and joints. Additionally, 39 markers were placed on the human dynamics model based on a Plug-in Gait full-body model.

#### 4.3.4. Inverse Kinematics Analysis

The variables in Equations (13) and (14) could be calculated using the marker positions attached to the driver, which were obtained from the joint moment estimation system and the human dynamics model obtained from the analysis in Section 4.3.3. The DhaibaWorks software was used for the computational process.

#### 4.3.5. Inverse Dynamics Analysis

The joint moments were estimated using Equations (13) and (14) in Section 4.2, external forces acting on the driver obtained in Section 4.3.2, human dynamics model obtained in Section 4.3.3, and human posture-dependent variables obtained in Section 4.3.4. The calculations were performed using the DhaibaWorks software.

#### 4.3.6. Calculation of Offset Values for Joint Moments

We calculated the change in the estimated joint moment applied to the driver under each experimental condition by considering an offset with the estimated joint moment when the driver maintained a stationary state, rather than using the joint moment estimates obtained by the process described in Section 4.3.5.

### 4.4. Joint Moment Estimation Results

Figure 21 and Figure 22 show examples of the simulation results of the ankle and knee joint moments under the following conditions: velocity of 2 km/h, turning radius of 0.5 m, and right turning direction. Figure 21 shows the results for manual turning, and Figure 22 for automatic turning. The red line indicates the right leg data, and the blue line indicates the left leg data.

First, we examined the extent to which the load on the ankle varied depending on the driving mode, separately for the legs inside and outside of the turn direction. Figure 23 shows the average values of the ankle joint moment (*z*-axis) during turning for each driving style. The ankle joint moment (*z*-axis) refers to the joint moment applied around the *z*-axis of the ankle in the human model, as shown in Figure 20. The direction of the moment applied when the body was tilted inward was defined as positive. Blue indicates manual turning and red indicates automatic turning. The results show that the moment applied to the leg inside the turn direction was 1.775 Nm (SE = 0.047) during manual driving and 2.387 Nm (SE = 0.037) during automatic driving. The values for the foot outside the turn direction are 3.162 Nm (SE = 0.046) for manual driving and 3.718 Nm (SE = 0.038) for automatic driving. The results of a paired-sample *t*-test show a significant difference (*) in the mean values of joint moments between the ankles inside and outside the turn directions, regardless of the driving style (manual driving, t (13653) = 32.416, *p* < 0.01) (automatic driving, t (17138) = 32.118, *p* < 0.01).

Furthermore, the extent to which the load on the ankle varied depending on foot position during turning was examined for each driving mode. The results of Welch’s *t*-test show a significant difference (**) in the mean values of joint moments between manual and automatic turning, regardless of foot position during turning (inward turning, t (27337) = 10.231, *p* < 0.01) (outward turning, t (28331) = 9.330, *p* < 0.01).

We also examined the extent to which the load on the knee varied depending on the driving mode, separately for the legs inside and outside of the turn direction. Figure 24 shows the average values of the driver’s knee joint moment (*z*-axis) during turning for each driving mode. The knee joint moment (*z*-axis) refers to the joint moment applied around the *z*-axis of the knee in the human model, as shown in Figure 20. The direction of the moment applied when the body was tilted inward of the turn was considered positive. Blue indicates manual turning and red indicates automatic turning. The results of Welch’s *t*-test indicate that the mean knee joint moment was significantly higher (**) in automatic driving than in manual driving (leg inside of the turn direction, t (27228) = 8.033, *p* < 0.01) (leg outside of the turn direction, t (27892) = 8.883, *p* < 0. 01).

Furthermore, we examined the extent to which the load on the knee varied during turning in driving mode. The results of a paired-sample *t*-test show that regardless of the driving mode, the mean value of the joint moments was significantly greater (*) for the knee outside of the turn direction than that inside of the turn direction (manual driving, t (13653) = 59.002, *p* < 0.01) (automatic driving, t (17138) = 60.063, *p* < 0.01).

These results indicate that when the driver turns in an IPV, the load on the legs, including the ankles and knees, is greater in automatic driving than in manual driving, and greater on the leg outside of the turn direction than on that inside. This is because the driver is more susceptible to centrifugal force during automatic turning than during manual turning. Thus, to withstand the centrifugal force generated during turning, it is easier for the driver to maintain balance by stepping harder on the outer leg than on the inner leg.

## 5. Conclusions

The dynamic behaviors of the drivers of IPVs during manual and automatic turning were investigated. The drivers moved the COG backward during the turning and deceleration of the vehicle. It was also found that under automatic turning conditions, the driver’s COG shifted in the opposite direction to the turning direction because of the centrifugal force acting on the driver. When the centrifugal force acting on the vehicle exceeds approximately 50 N during automatic turning, there is a risk of falling. As for the COP, it was found that balance during turning was maintained by moving the inner leg inward and the outer leg outward.

The load on the leg was evaluated by estimating the joint moments of the ankles and knees. The joint moments were larger in the outer leg than in the inner leg. The outer leg is considered to contribute to the postural stability during turning. The ankle and knee joint moments were significantly greater during automatic turning than during manual turning. In automatic turning, the driver exerts large joint moments to withstand unexpected centrifugal forces.

These findings indicate that to realize the automatic turning of an IPV, a control system is required that widens the distance between the COPs of the left and right legs during turning. Moreover, it should shift the COG inward and guide the vehicle into a posture that reduces the joint moments applied to both legs by reducing the effect of the centrifugal forces.

This study has several limitations. The drivers in the derivation experiment for COG/COP were three healthy young men, and those in the derivation experiment for the joint moment were five healthy young men. Although the experimental results showed significant differences, the number of subjects was small and limited to young individuals. Therefore, increasing the number of participants in the experiment and broadening the age range could be the focus of future research. Moreover, based on the findings of this study, we propose an automatic control method or a new mechanism for moving the COG of a driver, such as in manual turning, as a future research direction.

## Figures and Tables

**Figure 1 sensors-22-09931-f001:**
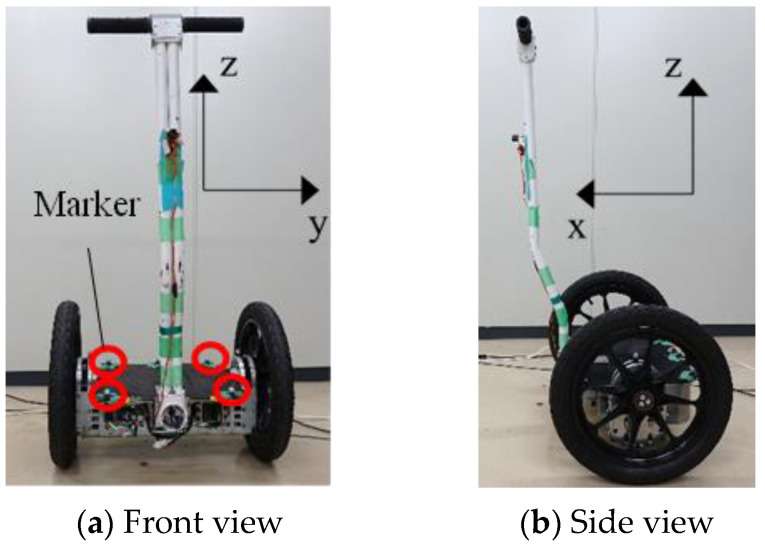
Experimental vehicle.

**Figure 2 sensors-22-09931-f002:**
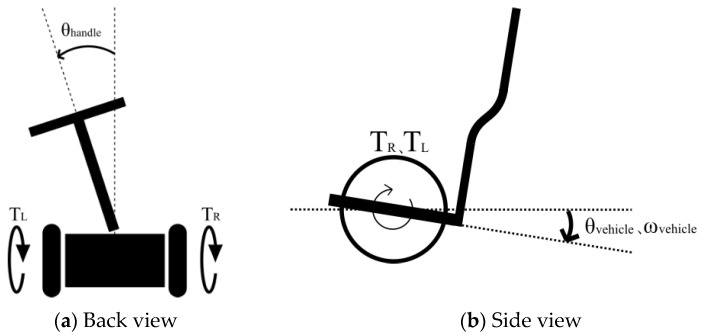
Sign definitions of $T_{R}$, $T_{L}$, $\theta_{vehicle}$, $\omega_{vehicle}$, and $\theta_{handle}$.

**Figure 3 sensors-22-09931-f003:**
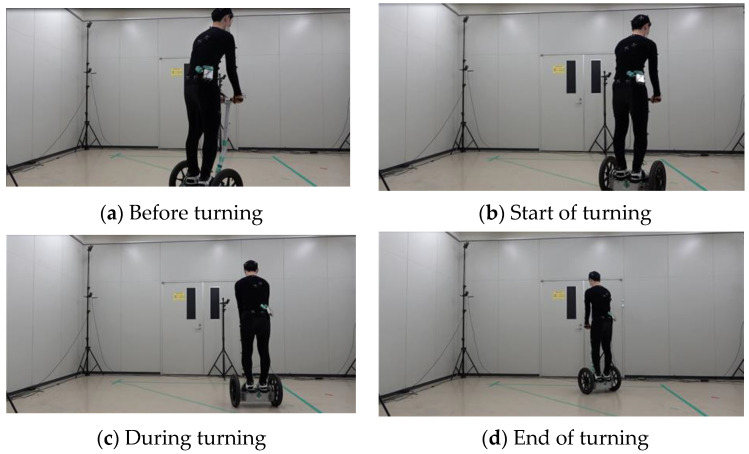
Turning experiment.

**Figure 4 sensors-22-09931-f004:**
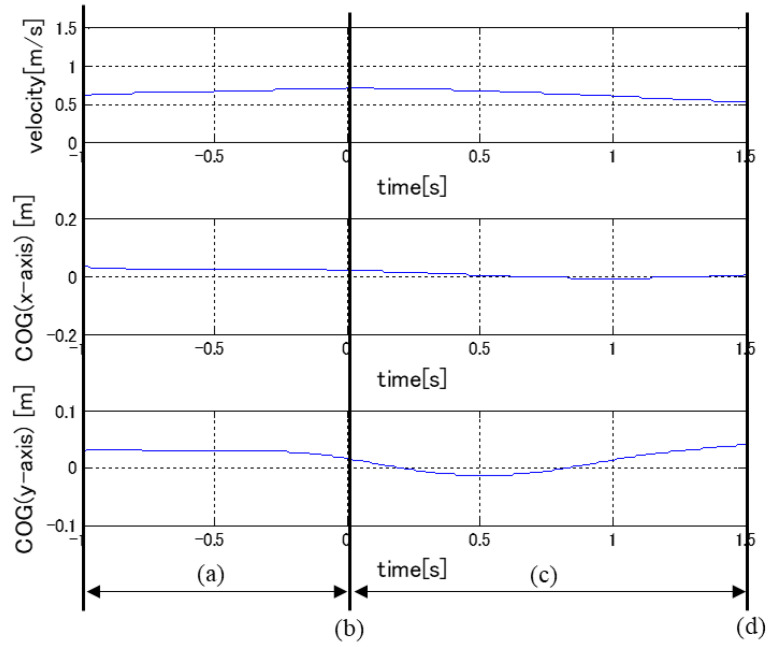
Example of experimental data. (a) Before turning, (b) Start of turning, (c) During turning, (d) End of turning.

**Figure 5 sensors-22-09931-f005:**
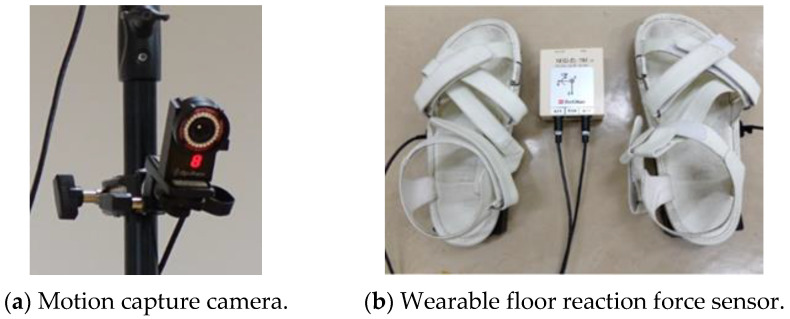
Experimental equipment.

**Figure 6 sensors-22-09931-f006:**
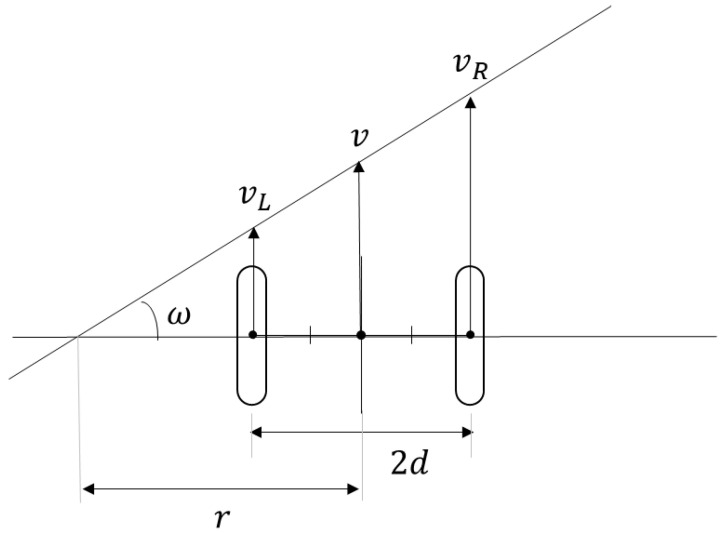
Schematic diagram of a vehicle in a left turn.

**Figure 7 sensors-22-09931-f007:**
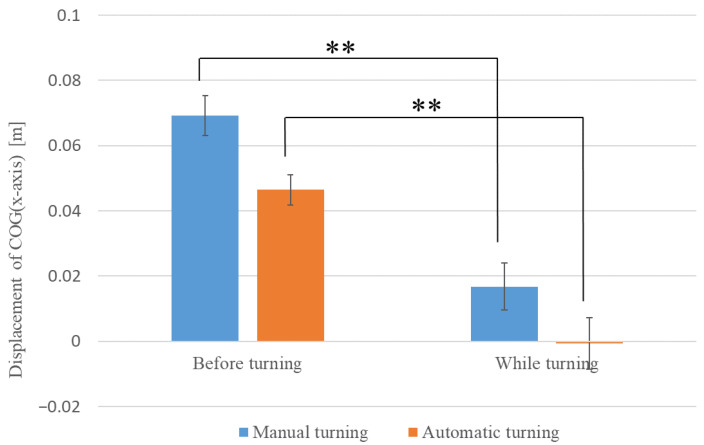
Average values of COG (*x*-axis) before and after turning.

**Figure 8 sensors-22-09931-f008:**
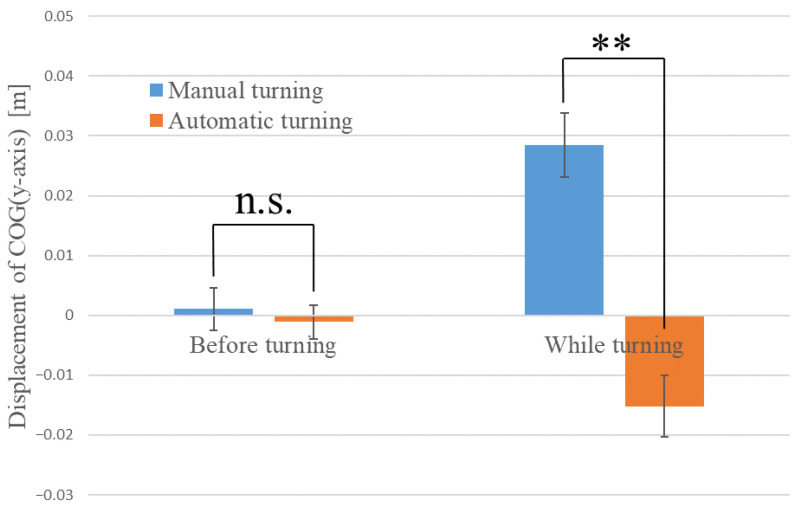
Average values of COG (*y*-axis) before and after turning.

**Figure 9 sensors-22-09931-f009:**
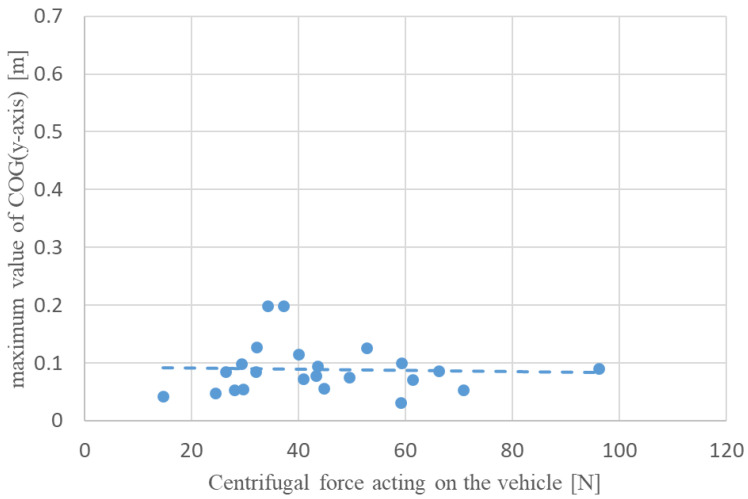
Relationship between the maximum value of COG (*y*-axis) and centrifugal force on the vehicle during manual turning.

**Figure 10 sensors-22-09931-f010:**
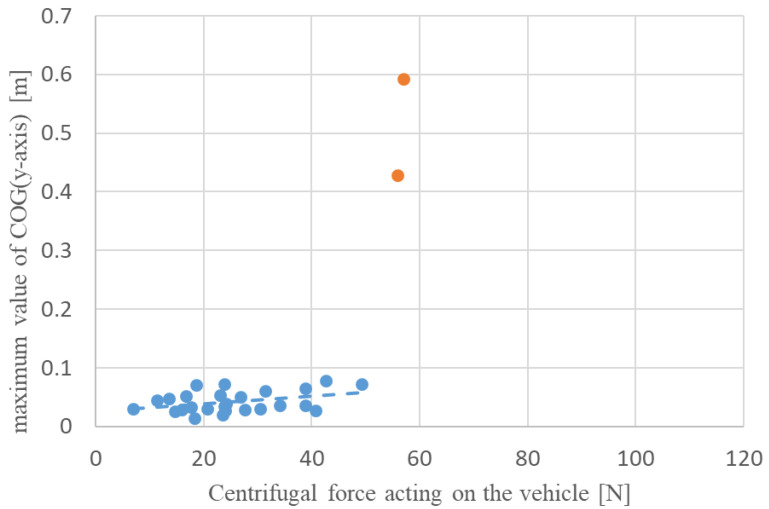
Relationship between the maximum value of COG (*y*-axis) and centrifugal force on the vehicle during automatic turning.

**Figure 11 sensors-22-09931-f011:**
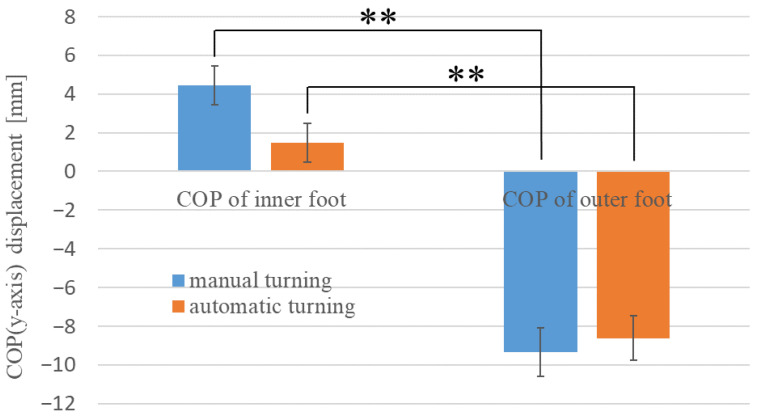
Average of COP (*y*-axis) during turning.

**Figure 12 sensors-22-09931-f012:**
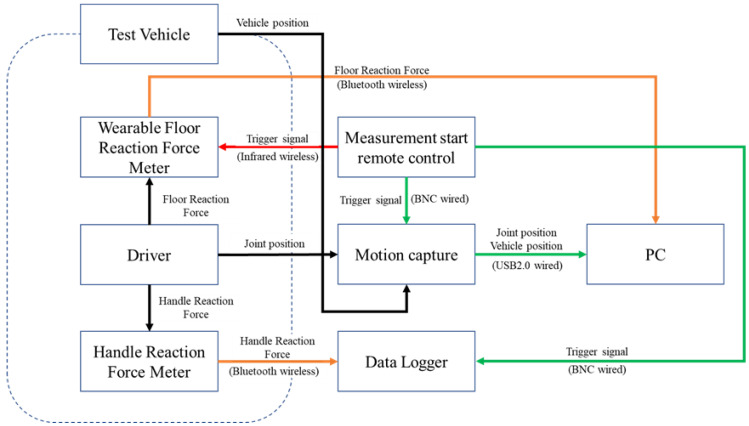
Flow diagram of the experimental setup.

**Figure 13 sensors-22-09931-f013:**
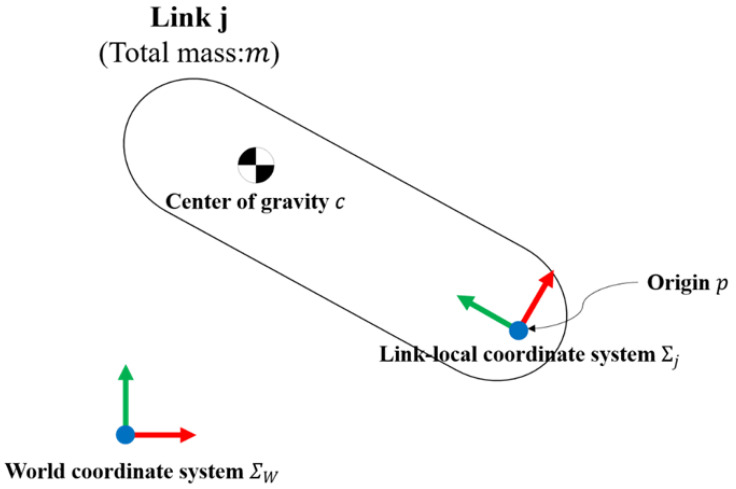
Schematic of a rigid link.

**Figure 14 sensors-22-09931-f014:**
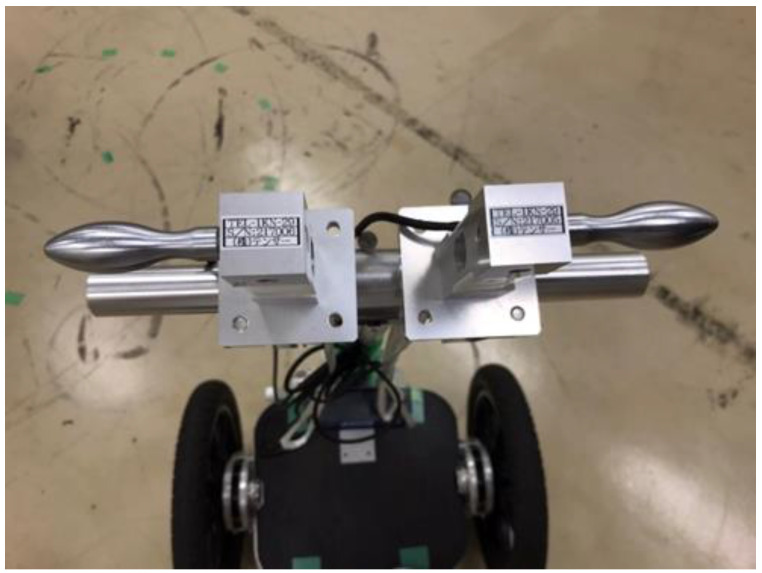
Handle reaction force meter.

**Figure 15 sensors-22-09931-f015:**
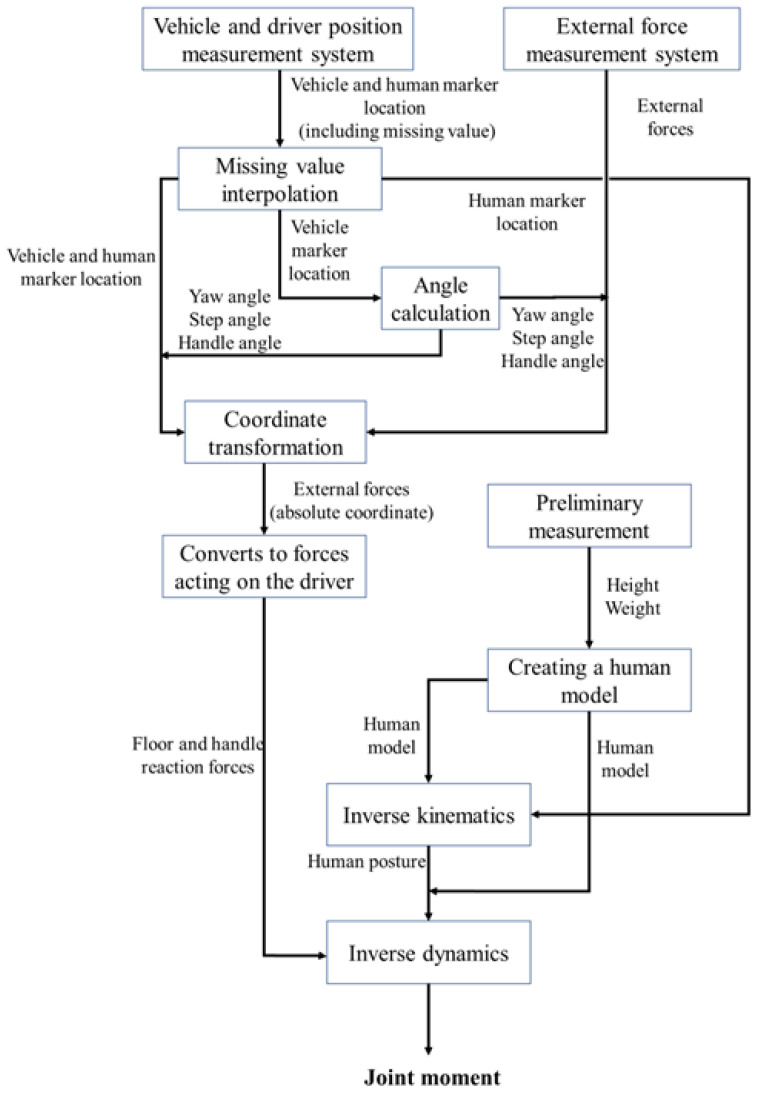
Data processing flow for determining joint moments.

**Figure 16 sensors-22-09931-f016:**
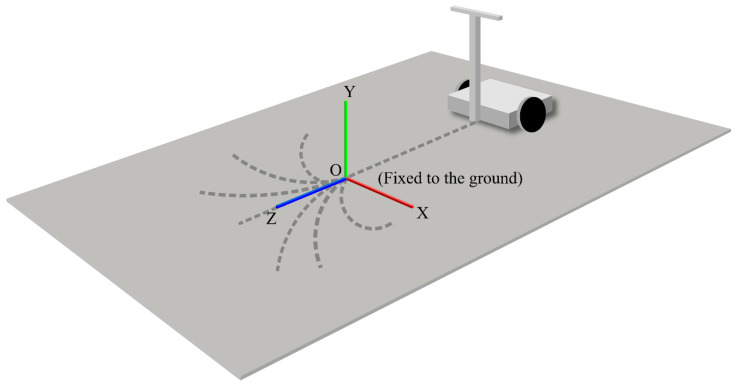
Absolute coordinate system.

**Figure 17 sensors-22-09931-f017:**
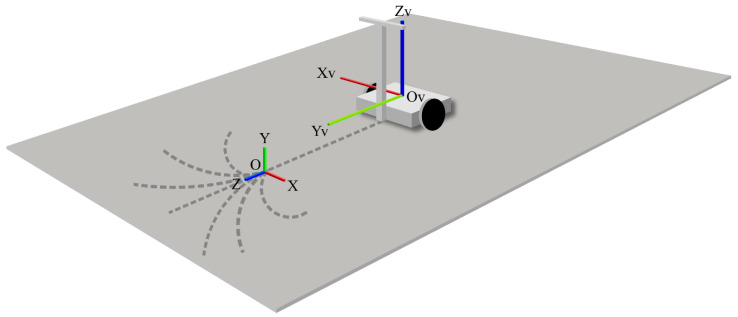
Vehicle coordinate system.

**Figure 18 sensors-22-09931-f018:**
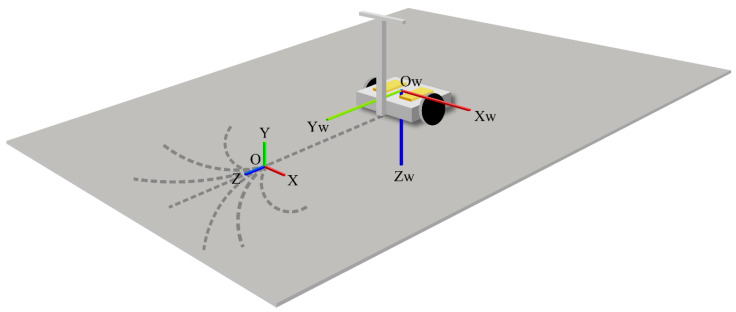
Coordinate system of the wearable floor reaction force meter.

**Figure 19 sensors-22-09931-f019:**
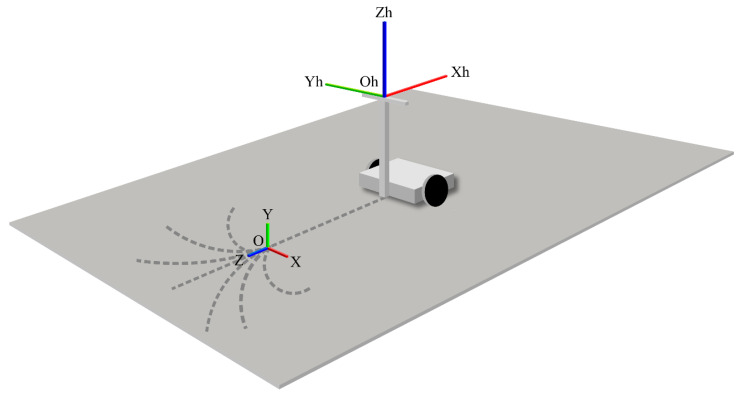
Coordinate system of the handle reaction force meter.

**Figure 20 sensors-22-09931-f020:**
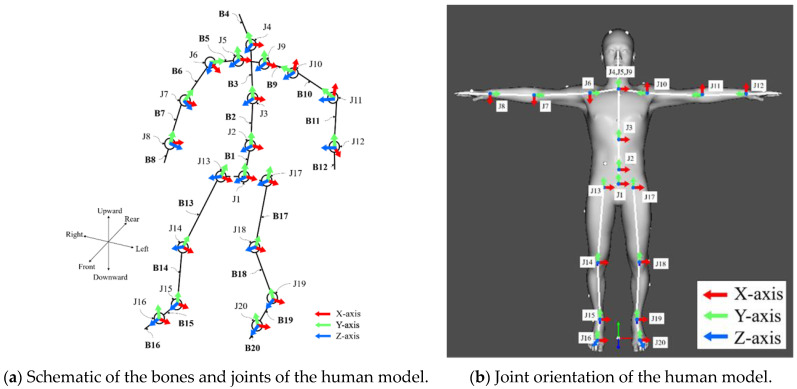
Composition of the human model.

**Figure 21 sensors-22-09931-f021:**
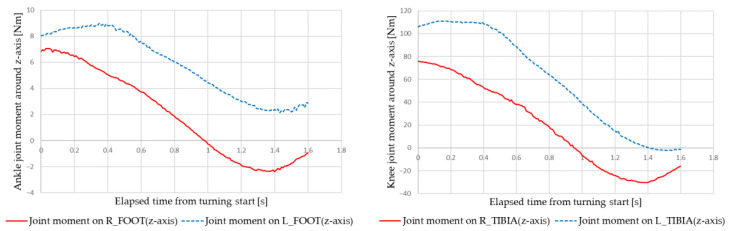
Ankle and knee joint moments during manual turning.

**Figure 22 sensors-22-09931-f022:**
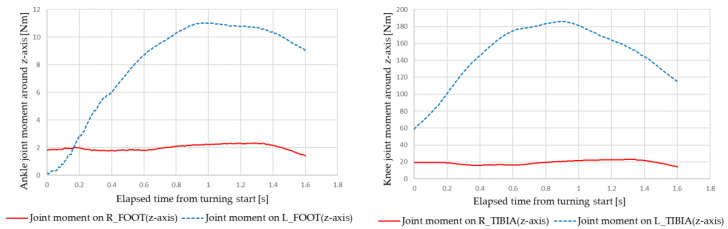
Ankle and knee joint moments during automatic turning.

**Figure 23 sensors-22-09931-f023:**
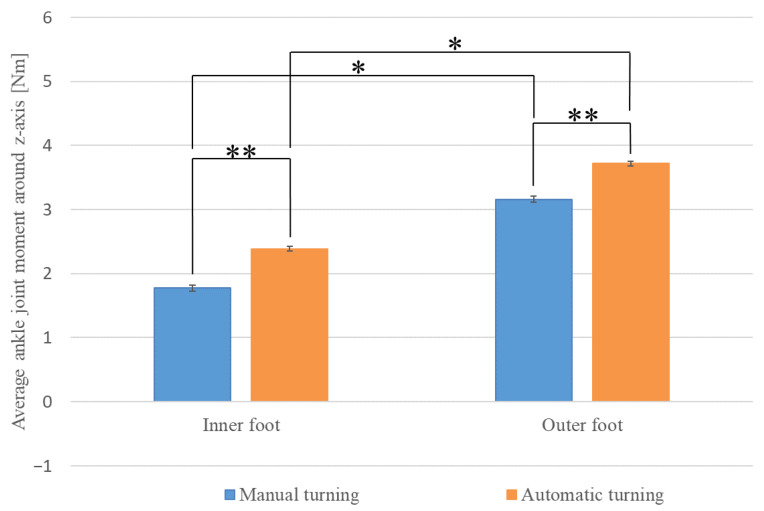
Average ankle joint moment around *z*-axis during turning.

**Figure 24 sensors-22-09931-f024:**
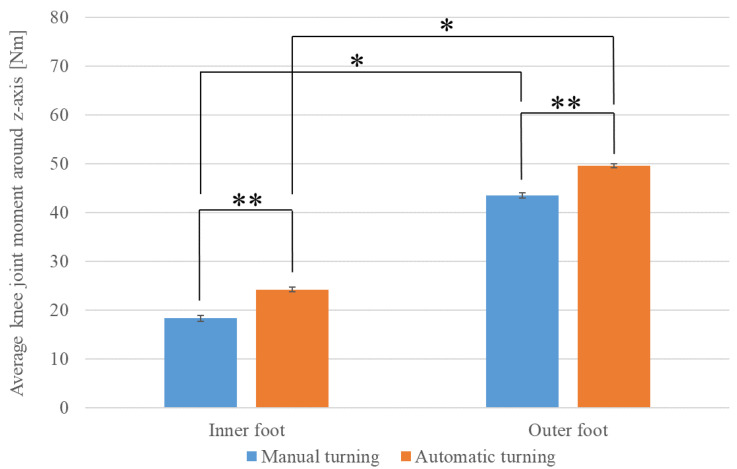
Average knee joint moment around *z*-axis during turning.

**Table 1 sensors-22-09931-t001:** Specifications of the experimental vehicle.

Definition	Value
Maximum velocity	10 km/h
Cruising range	10–15 km
Mass	30 kg
Length × Width	45 cm × 52.5 cm
Limitation of mass	20–80 kg
Output of motor	450 × 2 W

**Table 2 sensors-22-09931-t002:** Feedback gains for the experiments.

	$K_{1}$	$K_{2}$	$K_{3}$
Feedback gains	32	6.6	1.0

**Table 3 sensors-22-09931-t003:** Classification of correlation strength.

Correlation Coefficient r	Correlation Strength
$0.7\leq \left|r\right|$	High
$0.4\leq \left|r\right|<0.7$	Moderate
$0.2\leq \left|r\right|<0.4$	Low
$\left|r\right|<0.2$	Slight

**Table 4 sensors-22-09931-t004:** List of bone names.

Bone Index	Bone Name	Bone Index	Bone Name
B1	PELVIS	B11	L_ULNA
B2	SPINE	B12	L_HAND
B3	STERNUM	B13	R_FEMUR
B4	NECK	B14	R_TIBIA
B5	R_CLAVICLE	B15	R_FOOT
B6	R_HUMERUS	B16	R_TOE
B7	R_ULNA	B17	L_FEMUR
B8	R_HAND	B18	L_TIBIA
B9	L_CLAVICLE	B19	L_FOOT
B10	L_HUMERUS	B20	L_TOE

**Table 5 sensors-22-09931-t005:** List of joint names.

Bone Index	Bone Name	Bone Index	Bone Name
J1	Joint of PELVIS	J11	Joint of L_ULNA
J2	Joint of SPINE	J12	Joint of L_HAND
J3	Joint of STERNUM	J13	Joint of R_FEMUR
J4	Joint of NECK	J14	Joint of R_TIBIA
J5	Joint of R_CLAVICLE	J15	Joint of R_FOOT
J6	Joint of R_HUMERUS	J16	Joint of R_TOE
J7	Joint of R_ULNA	J17	Joint of L_FEMUR
J8	Joint of R_HAND	J18	Joint of L_TIBIA
J9	Joint of L_CLAVICLE	J19	Joint of L_FOOT
J10	Joint of L_HUMERUS	J20	Joint of L_TOE

## Data Availability

The data presented in this study are available upon request from the corresponding author.

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
