# Peer review of "Differences in Driver Behavior between Manual and Automatic Turning of an Inverted Pendulum Vehicle"

_sensors, 2022, doi:10.3390/s22249931_

Round 1

Reviewer 1 Report

 This paper has proposed the driver's behavior during turning is experimentally investigated. The displacement of the driver's COG was found to move backward during the turn and decelerate the vehicle. As for COP, it was found that the driver maintained balance by moving the inner foot more inward and the outer foot more out- ward during the turn. However, this paper does not write well and is not acceptable to publish in its current form., the reviewer will provide several comments that should be considered carefully as follows.

-The Introduction section does write well. Please add your motivation, contribution and outline paragraphs.

-Where is the related work section?

-How is your work deference compared with others?

-Some relevant studies should be reviewed as follows. (1) Provably Secure with Efficient Data Sharing Scheme for Fifth-Generation (5G)-Enabled Vehicular Networks without Road-Side Unit (RSU); (2) CM-CPPA: Chaotic Map-Based Conditional Privacy-Preserving Authentication Scheme in 5G-Enabled Vehicular Networks; (3) Chebyshev Polynomial-Based Scheme for Resisting Side-Channel Attacks in 5G-Enabled Vehicular Networks

-What is the security model? System model? Security requirement definition? Please add them as a new section called background before proposing the section.

-Please add the limitation of this paper and future work at end of the conclusion section.

-please use the LATEX (overleaf) instead of the word version.

-All references are old.

Reviewer 2 Report

1-Equation 1 and 2 are not defined. All the terms and variables needs to be defined.

2-The number of equation should be after the equation appears. This should be followed for all the equations.

3-The references of equation 1 and 2 are not given. SImilarly, the rest of equations also miss the references. It should be clearly listed what is from the literature and what has been proposed.

4-k1, k2, k3 and alpha are not defined. What exactly these parameters depend on and reference.

5-The introduction is short. New references on the driver behavior are needed to be discussed. For example,

A New Driver Model Based on Driver Response F Ali, ZH Khan, FA Khan, KS Khattak, TA Gulliver - Applied Sciences, 2022     

6-References of equations 7 and 8 are not given.

7-The simulations graphs are not given. A few graphs should be shared to clearly understand the dynamics behavior.

8-The structure of the journal needs to be revised. It is confusing.

9-English and grammatical mistakes exist and needs to be clearly revised.

10-Very few latest references are given. Atleast 5 references from 2022 needs to be discussed so that the literature is uptodate.

11-Conclusion is missing information. It needs to be revised. 12-The Abstract is not concrete and concise.  Experimental setup flow diagram should be given.  The methodology should be given and discussed.  

Round 2

Reviewer 1 Report

Authors pay extra attention to address reviewers' concerns. Done have another issue. My decision for this paper is accepted.

Reviewer 2 Report

All the comments are addressed and can be published in the current form.